# A Comparison of the Effects of Young-Child Formulas and Cow’s Milk on Nutrient Intakes in Polish Children Aged 13–24 Months

**DOI:** 10.3390/nu13082511

**Published:** 2021-07-23

**Authors:** Malgorzata Kostecka, Izabella Jackowska, Julianna Kostecka

**Affiliations:** 1Faculty of Food Science and Biotechnology, University of Life Sciences in Lublin, Akademicka 15, 20-950 Lublin, Poland; izabella.jackowska@up.lublin.pl; 2Faculty of Medicine, Medical University of Lublin, Chodźki 19, 20-093 Lublin, Poland; kostecka.julianna@gmail.com

**Keywords:** young-child formula, cow’s milk, nutrient intake, food ration model, children

## Abstract

Adequately balanced daily food rations that provide the body with sufficient amounts of energy and nutrients, including minerals, are particularly important in early childhood when rapid physical, intellectual and motor development takes place. Cow’s milk (CM) and young-child formulas (YCFs) are introduced to a child’s diet past the first year of age. The main aim of the present study was to perform a qualitative and a quantitative analysis of daily food rations of young children based on the recommendations of the daily food ration model. An attempt was also made to determine whether the type of consumed milk (YCF or CM) adequately meets young children’s energy demands and contributes to the incorporation of different food groups into a balanced and healthy diet for children aged 13–24 months. A total of 714 parents between October 2019 and March 2020 filled out a food frequency questionnaire. In the second stage of the study, the parents participated in a dietary recall and were asked to keep diaries of all meals and foods consumed by children over a period of three days. The mean daily intake of CM/YCF and fermented milks was determined at 360 mL ± 128 mL, and it accounted for 55.4% of the guideline values. Flavored dairy products were consumed more frequently than fermented milks without added sugar or flavoring (94 ± 17 g vs. 56 ± 26 g, *p* < 0.05). Diets incorporating CM were significantly more abundant in protein than YCF diets (29.3 g vs. 21.9 g; *p* < 0.01). Liquid intake was somewhat higher in children fed YCFs (1280.8 mL vs. 1120.1; *p* < 0.05), mainly due to the higher consumption of fruit juice, nectars and sweetened hot beverages (246 ± 35 mL in the YCF group vs. 201 ± 56 mL in the CM group; *p* < 0.05). Children fed YCF consumed significantly larger amounts of sweetened beverages such as tea sweetened with sugar or honey, sweetened hot chocolate or instant teas (OR = 2.54; Cl: 1.32–3.26; *p* < 0.001), than children receiving CM. This group was also characterized by higher consumption of sweetened dairy products, mainly cream cheese desserts, fruit yogurt and yogurt with cereal (OR = 1.87; Cl: 1.36–2.54; *p* < 0.01), as well as a lower daily intake of plain fermented milks (OR = 0.56; Cl: 0.21–0.79; *p* < 0.001). The daily food intake and the quality of the diets administered to children aged 13–24 months were evaluated and compared with the model food ration. It was found that milk type influenced children’s eating habits and preference for sweet-tasting foods. The study also demonstrated that Polish parents and caregivers only have limited knowledge of nutritional guidelines for toddlers.

## 1. Introduction

Adequately balanced daily food rations that provide the body with sufficient amounts of energy and nutrients, including minerals, are particularly important in early childhood when rapid physical, intellectual and motor development takes place [1]. Cow’s milk (CM) and young-child formulas (YCFs), also referred to as follow-up formulas [2] or growing-up milk (GUM) [3], are introduced to a child’s diet past the first year of age. The term “young-child formula” has been recommended by the European Food Safety Authority (EFSA) because it most accurately describes the age group for which it is intended (1 to 3 years). Not all YCFs contain animal protein, which is why “formula” is preferred over “milk”. The term “growing-up” is not recommended because it implies that the formula exerts a specific influence on growth. In an attempt to standardize the applicable terminology, the Committee on Nutrition (CoN) of the European Society for Pediatric Gastroenterology, Hepatology and Nutrition (ESPGHAN) has also recommended “young-child formula” as the most appropriate term [4]. Young-child formulas have a highly varied composition that is not legally regulated, and they may be abundant in protein, carbohydrates as well as large amounts of added sugar in the form of sucrose. According to ESPGHAN guidelines, the protein content of YCFs should approximate the lower threshold value. Research has demonstrated that YCFs increase the intake of vitamin D, iron and n-3 PUFAs. However, these nutrients can be also supplied with unprocessed and/or fortified foods. Based on the existing evidence, ESPGHAN has concluded that there are no grounds for routinely administering YCFs to children aged 1 to 3 years, but YCFs can be a useful strategy to compensate for nutritional deficiencies in children’s diets [2,4,5].

Cow’s milk is widely consumed by young children in some countries, including Mexico and the United States. In Mexico, more than 86% of 1- to 3-year-olds consume milk daily [6], and 10%–13% of their daily energy intake comes from milk [7]. In turn, only 4%–5% of the daily energy intake of Chinese children comes from CM, and this product is rarely incorporated in child diets, in particular in the diets of children aged 1 to 2 years [8,9].

Several studies have been conducted to compare the nutrient intakes of children consuming YCFs and CM [10,11,12]. A report from Germany described the similarities and differences between the contribution of 200 mL of YCF and 200 mL of CM (1.5% fat) to the recommended intakes of energy and macro- and micronutrients [13]. However, energy and nutrient intakes during infants’ transition from milk to solid foods have been rarely described comprehensively in the literature [14,15], and there is also a general scarcity of recent studies for the population of Polish toddlers [16].

The daily food ration model for young children developed by Weker et al. [17] is based on Polish norms. The model supports meal planning based on a child’s specific energy and nutrient requirements, and it covers six groups of food products.

The main aim of the present study was to perform a qualitative and a quantitative analysis of daily food rations of young children based on the recommendations of the daily food ration model. An attempt was also made to determine whether the type of consumed milk (YCF or CM) adequately meets young children’s energy demands and contributes to the incorporation of different food groups into a balanced and healthy diet for children aged 13–24 months.

## 2. Methods

### 2.1. Study Sample

The parents of children aged 13–24 months who had introduced CM to their children’s diets at one year of age or continued to administer modified milk past the first birthday were invited to participate in the study. The exclusion criteria were allergy or the risk of allergy (which would require the use of extensively hydrolyzed formulas or amino acid formulas) and other metabolic diseases that require specialist foods. All of the evaluated children were full-term babies with birth weights higher than 2500 g. None of the children was breast-fed at the time of the study.

### 2.2. Data Acquisition

The study was conducted in two stages, but not all parents who participated in the first stage took part in the second stage of the research.

A total of 714 parents whose children were patients of pediatric clinics in Lublin, Kraśnik and Rzeszów in southeastern Poland between October 2019 and March 2020 consented to participate in the study. In the first stage, the participants filled out a food frequency questionnaire designed based on a validated KomPAN questionnaire [18,19]. During the first meeting, the parents were asked to complete a paper questionnaire containing 25 questions on the frequency of consumption of various food groups, serving size and daily intake of CM or modified milk, formula brand incorporated in the diet, factors influencing the parents’ choice of formula, dietary supplements and fortified foods, as well as basic information about the child’s age, body weight and height. A total of 592 correctly completed questionnaires were returned. In the second stage of the study, the parents participated in a dietary recall and were asked to keep diaries of all meals and foods consumed by children over a period of three days. A dedicated form was provided for the purpose, and food dairies were to be completed in the following week. A researcher/student/nurse instructed the participants on how to keep the diary and answered the parents’ questions. The parents were asked to record all meals and foods consumed by the child over a period of three non-consecutive days, including Saturday or Sunday. Caregivers were instructed to note the time of each meal, the weight (or volume) of all ingested foods and to accurately describe all food products (including brands), food preparation methods and detailed recipes for all foods prepared at home. Serving size was determined with the use of household measures and utensils (slice, spoon, cup, etc.) and based on photographs of food products and dishes that were presented to the parents during the questionnaire completion.

Food diaries were filled out correctly by 486 parents, and they were divided into three groups:

Group 1—YCF group comprising 178 children who were fed modified milk with a minimum daily intake of 250 mL;

Group 2—CM group comprising 156 children who were fed CM past the age of 12 months with a minimum daily intake of 250 mL, but did not receive YCF or dairy products based on YCF;

Group 3—mixed group of 152 children who were fed both CM and modified milk intended for children aged 1–3 years.

### 2.3. Analysis of Food Dairies

The results of the three-day dietary recall were processed in the Dieta 6 program developed by the Independent Unit of Nutritional Epidemiology and Recommended Dietary Intakes of the National Food and Nutrition Institute in Warsaw based on Polish norms [20,21]. The program lists the composition of around 3000 foodstuffs that are widely used for culinary purposes in Polish households, as well as commercial food products for infants and young children (under 3 years of age) available on the Polish market. The composition of less than 12 homemade foods could not be determined, and it was inferred based on similar products. The composition of all baby foods was stated on the nutrition facts label. Energy and nutrient intakes were calculated for each child based on the total food intake during the three-day dietary recall. In addition to total energy intake, the following nutrients were also considered in the analysis: proteins; lipids (with a division into saturated, monounsaturated and polyunsaturated fatty acids); total carbohydrates, including sucrose and lactose (excluding fiber); minerals Na, K, I, Ca, P, Mg, Zn and Fe; water-soluble vitamins B1, B2, B6, B12, C and folates; and fat-soluble vitamins E and D (only vitamins occurring naturally in foods) and total vitamin A. In Poland, children aged 1 to 2 years are frequently prescribed vitamin D and multivitamin supplements. These supplementary intakes were not taken into account in the current study because they are not used regularly throughout the year and are not recommended by pediatricians.

### 2.4. Statistical Analysis

The mean intake of each nutrient in each group was calculated. The mean intake of each nutrient in each group was compared against the dietary guidelines for CM and YCF in the independent *t*-test (parametric data) and the non-parametric Kruskal–Wallis test. Mean nutrient intakes were compared between the CM group and the mixed group, and between the YCF group and the mixed group in Dunn’s post hoc test.

Adequate intake (AI) values were applied to nutrients without established recommended daily allowance (RDA) cut-offs, including vitamins D and E. The RDA values for protein, calcium, phosphorus, magnesium, iron, selenium, zinc, copper, thiamin, riboflavin, vitamins B6 and B12, folate and vitamin C and AI values for sodium and potassium were determined based on Polish Dietary Guidelines (National Food and Nutrition Institute, 2012). In an analysis of macronutrients (excluding protein and carbohydrates for which RDA values are available), the energy intake from fat was determined based on the acceptable macronutrient distribution ranges (AMDR). The recommended fiber intake was established based on AI values.

The normality of distribution of continuous variables was assessed in the Shapiro–Wilk test. For continuous variables, data were presented as means with a 95% confidence interval (95% CI). Three groups of respondents were analyzed as follows: (i) CM group, (ii) YCF group and (iii) mixed group. The significance of odds ratios (ORs) was assessed with Wald’s statistics. 

Data were processed in the Statistica program (version 12.0 PL, StatSoft Inc., Tulsa, OK, USA, StatSoft, Krakow, Poland) at three levels of significance (*p*): *p* < 0.05, <0.01 and <0.001.

## 3. Results

### 3.1. Analysis of Daily Food Rations

The estimated number of servings and serving size of different food products in the daily rations of the evaluated children differed from the guideline values (Table 1).

The evaluated children consumed more protein foods, in particular meat, cold cuts and processed meat, which increased the percentage of protein foods in the daily ration (Table 2). The consumption of dairy products was below the recommended levels. The mean daily intake of CM/YCF and fermented milks was determined at 360 mL ± 128 mL, and it accounted for 55.4% of the guideline values. Flavored dairy products were consumed more frequently than fermented milks without added sugar or flavoring (94 ± 17 g vs. 56 ± 26 g, *p* < 0.05). Children’s diets were more abundant in wheat and whole wheat (graham) bread than whole grain bread (*p* < 0.05), and highly processed breakfast cereals with added sugar were consumed more frequently than oatmeal or buckwheat groats (27 ± 12 g vs 10 ± 10 g; *p* < 0.05). The analyzed diets were deficient in vegetables and fruit, and children consumed 2–3 servings of vegetables and fruit daily on average. Most diets were monotonous, and they featured mostly orange, red and green vegetables, whose daily consumption reached 140 ± 59 g vs. the recommended amount of 200 g. Carrots, tomatoes and lettuce were most frequently consumed, whereas cruciferous vegetables, including cabbage and cauliflower, as well as pumpkin, were the least popular. Daily consumption of fruit, mostly yellow and orange fruit (apples and bananas), accounted for 86% of the recommended daily intake. Average juice consumption was determined at 220 ± 29 mL, and it exceeded the recommended daily amount of 125 mL. The consumption of freshly squeezed fruit juice, pasteurized fruit juice, bottled fruit nectars and juice was too excessive (135 ± 15 mL vs. 175 ± 35 mL vs. 200 ± 25 mL). The consumption of sugar and sweets differed across the analyzed diets (*p* < 0.05). In the model food ration, daily sugar intake is limited to 20 g. Only 24% of the children consumed 20–24 g of sugar daily (100%–115% of the guideline values). In 11% of the cases, daily sugar intake was determined at 16–19 g and was below guideline values. However, in 65% of the examined diets, daily sugar intake exceeded the guideline levels 2- to 3-fold. These children consumed 53 ± 9 g sugar on average, mostly from sweetened dairy products such as cream cheese desserts, as well as candy bars, chocolate, gummy sweets and sweetened juice.

### 3.2. The Influence of Young-Child Formulas and Cow’s Milk on the Nutritional Value of the Diets of Children Aged 13–24 Months

The nutritional composition of YCFs available on the Polish market (six brands, for children aged 1 year and older) and the average composition of whole CM are shown in Table 3. The remaining YCFs were characterized by a higher content of saturated fatty acids (1.33 g/100 mL) that was similar to CM, as well as high sucrose content (1.2–2.4 g/100 mL).

All YCF brands contained dietary fiber (galacto-oligosaccharides/fructo-oligosaccharides). The greatest differences were observed in the content of iron, vitamins D and C and folic acid, which was very low in (unfortified) CM and high in YCF. One YCF brand was additionally fortified with calcium (217 mg/100 mL), magnesium (20 mg/100 mL) and iodine (42 µg/100 mL), and it was significantly more abundant in iodine than the remaining YCFs, where the average iodine content was determined at 18.03 ± 6.01 µg/100 mL.

The mean daily intake of YCF was higher (283 mL) than the intake of CM (261 mL), but the difference was not statistically significant. The average nutritional value of YCF diets was significantly (*p* < 0.05) higher than that of CM diets (Table 4). Diets incorporating CM were significantly more abundant in protein than YCF diets (29.3 g vs. 21.9 g; *p* < 0.01). Cow’s milk and natural dairy products were the main sources of dietary protein. Average carbohydrate intake in the analyzed population was similar to that of children fed CM, whereas the consumption of YCF increased the content of carbohydrates, in particular sucrose, in the daily ration (67.6 g vs. 33.8 g; *p* < 0.001).

Liquid intake was somewhat higher in children fed YCFs (1280.8 mL vs. 1120.1; *p* < 0.05), mainly due to the higher consumption of fruit juice, nectars and sweetened hot beverages (246 ± 35 mL in the YCF group vs. 201 ± 56 mL in the CM group; *p* < 0.05).

Daily rations were also compared for the content of vitamins and minerals (Table 5). Vitamin A content was elevated in all cases, in particular in the YCF group, where vitamin A intake accounted for 231.5% of the reference value (*p* < 0.001), which was attributed to the consumption of fortified YCFs, dairy products and carrots. In the CM group, the dietary intake of vitamin D was very low (2.11 µg, 42.2% of AI values), mainly because CM, unlike YCF, is not fortified with vitamin D.

The intake of water-soluble vitamins was equivalent to 86%–124% of the guideline values in diets containing both CM and YCF. Considerable differences were observed only in vitamin C intake, which reached 217% of the guideline levels in YCF diets (*p* < 0.01).

The analyzed diets met the children’s demand for calcium, iron and iodine. Phosphorus (166.4% in the YCF group vs. 134.7% in the CM group; *p* < 0.01) and sodium (135.2% in the YCF group vs. 126.2% in the CM group; *p* < 0.05) intake was very high. The recommended intake of potassium was exceeded in the YCF group (148.8% of the reference value) despite the fact that YCF preparations on the Polish market are less abundant in potassium than CM.

### 3.3. The Associations between the Type of Consumed Milk and the Composition of the Diets of Children Aged 13–24 Months

Food diaries and the results of dietary interviews with parents were analyzed to identify products that were most widely consumed by children fed CM and YCFs (Table 6).

Children aged 13–24 months fed YCF consumed significantly larger amounts of sweetened beverages such as tea sweetened with sugar or honey, sweetened hot chocolate or instant teas (OR = 2.54; Cl: 1.32–3.26; *p* < 0.001) than children receiving CM. This group was also characterized by higher consumption of sweetened dairy products, mainly cream cheese desserts, fruit yogurt and yogurt with cereal (OR = 1.87; Cl: 1.36–2.54; *p* < 0.01), and lower daily intake of plain fermented milks (OR = 0.56; Cl: 0.21–0.79; *p* < 0.001).

In the group of children fed CM, the consumption of plain fermented milks was significantly higher (OR = 1.84; Cl: 1.26–2.41; *p* < 0.01) and the daily intake of sweetened dairy products was lower (OR = 0.64; Cl: 0.26–0.89; *p* < 0.001) than in the YCF group. The mean daily intake of plain yogurt was determined at 74 ± 12 g in the CM group and at 39 ± 21 g in the YCF group. The frequency of consumption and the size of breakfast cereal portions were higher in the YCF group, but the difference was not statistically significant. The daily intake of sweets was also similar in the compared groups. The most popular sweets were chocolate, chocolate-filled milk bars and gummy sweets in the YCF group, and candy bars, ice cream and chocolate and hazelnut spreads in the CM group.

## 4. Discussion

In the present study, the daily food ration of toddlers (aged 13–24 months) was evaluated and compared with the model daily ration, and the influence of CM and YCFs on the mean daily nutrient intake was determined. Dietary behaviors and choices in early childhood play a significant role in healthy growth and development. Despite that, the quality of diets consumed by children younger than 24 months has been rarely analyzed in the literature [22,23].

### 4.1. Daily Food Ration

Based on the guidelines of the World Health Organization (WHO), daily consumption of added sugar should be less than 10% of total daily energy intake. Based on a limited number of population-wide studies of sugar intake, the WHO also concluded a study that a reduction in the intake of added sugar to below 5% of total energy intake would decrease the risk of dental caries in children [24]. According to the Scientific Advisory Committee on Nutrition (SACN) in the UK and the American Heart Association (AHA), daily sugar intake should be below 5% of total energy intake, which implies that children younger than 3 years should consume less than 3 teaspoons of sugar, which is equivalent to 170 mL of fruit nectar (81–260 mL) [25,26]. In the current study, daily sugar intake was equivalent to 135% (CM group) to 270% (YCF group) of the reference value, and the main sources of sugar were sweetened beverages, juice and commercially available complementary foods (CACFs). Almost 30.0% of CACFs (such as fruit juice and concentrates) in Austria, 32.6% of CACFs in Israel, 37.5% of CACFs in Hungary and 41.4% in Bulgaria contained added sugars and sweeteners [27]. In the USA, more than 70.0% of CACFs for toddlers contained one or more added sugars [28]. In the present study, 72% of children consumed at least one type of sweets daily. Significantly lower values were noted by Yu et al. (2016), where only 40% of children aged 21–35 months consumed one type of sweets such as biscuits, honey and chocolate [9]. In some studies, sugar intake was higher among girls than boys [29]. In contrast, Yuan et al. [30] reported that boys consumed more sugar and sweet snacks than girls at the age of 8 months. In the present study, the consumption of sugar and sweets did not differ between the sexes, and similar observations were made by Masztalerz-Kozubek (2020) in a study of Polish children aged 12–36 months [31].

Research conducted in the USA and Brazil demonstrated that the intake of fruit and vegetables was less than adequate among infants and young children [32,33,34,35]. Unlike in the present study, the Nestlé Feeding Infants and Toddlers Study (FITS) (2016) revealed that 27% of 2- to 3-year-olds did not consume the recommended amounts of vegetables [34]. In this study, orange vegetables were most popular, and up to 64% of the surveyed children consumed 1 portion of these products. In contrast, dark green vegetables were consumed by every fourth child, and red vegetables by every third child. Only 40% of the evaluated children consumed dark green leafy vegetables, and the consumption of deep yellow vegetables was even lower [9]. In a study by Deming et al. [36], fewer children consumed dark yellow than dark green vegetables. The selection of vegetables in the diet plays a very important role in child nutrition because brightly colored vegetables are more abundant in bioactive components. The content of antioxidants from vegetables and fruits is great importance for health, and diet plays a crucial role in the regulation of chronic inflammation [37,38].

Daily fruit intake was below the recommended level of 4 servings, and most daily rations contained 2–3 fruit servings. Studies analyzing the diets of infants and young children in 2002 and 2008 [33,34,39] demonstrated that one out of five children aged 9–12 months did not consume any fruit, whereas in children older than 12 months, the recommended daily fruit intake was met in only 50% of the cases. In a study by Yu [9], 38% of young children did not consume any fruit.

In the examined diets, daily milk intake was equivalent to 55.4% of the reference value, which is a reason for concern. Insufficient consumption, especially of cow’s milk, can lead to a decreased level of calcium. Milk and dairy are the main source of calcium, which stored in bones and teeth. Optimal peak bone mass and bone health later in life are favored by a sufficient calcium intake in infancy, childhood and adolescence. It is essential for prevents rickets and osteoporosis [40,41,42]. Lower milk consumption was accompanied by a higher intake of sweetened beverages and juice. In the work of Kay et al. [43], whole milk and 100% fruit juice were the most popular beverages among children aged 12–23.9 months. In older children (24–47.9 months), the consumption of fruit juice and sugar-sweetened beverages (SSB) was higher, and milk intake was lower [44].

### 4.2. Consumption of Cow’s Milk and Young-Child Formulas and Nutritional Requirements

Modified milk is a dairy product fortified with minerals and vitamins, which can compensate for nutrient deficiencies that are sometimes observed in children aged 12–24 months during the transition to solid foods [2,4,45]. There is limited research on the contribution of milk to the diets of children younger than 2 years [10,46,47]. Young-child formulas could cater to the special dietary requirements of undernourished children, but the resulting intake of protein, sodium and vitamin A can exceed the recommended levels, whereas the intake of docosahexaenoic acid (DHA), arachidonic acid (ARA) and vitamin D can be insufficient [45].

The examined diets were deficient in high-quality fats. The consumption of saturated fatty acids (SFAs) tended to be excessive, whereas the intake of polyunsaturated fatty acids (PUFAs) tended to be insufficient regardless of the type of consumed milk. Similar observations were made in Australia, New Zealand [11], Germany [48] and the USA [49], where SFAs from dairy products were consumed in excess, whereas the dietary intake of PUFAs was low, even among children receiving fortified YCF. The results of dietary recalls differed significantly from nutritional models, which demonstrated that the replacement of CM with YCF would increase the consumption of healthy fats and reduce the intake of SFAs [12,50].

Children in the second and third years of life are vulnerable to iron depletion, which may progress to iron-deficiency anemia (IDA) [51]. Iron depletion occurs in up to 30% of young children in developed countries [52,53]. In the present study, the mean daily intake of iron was determined at 7.45 ± 1.9 mg, and it was equivalent to 106% of the reference value. Diets combining CM and YCFs met the demand for iron in 100% and supplied adequate amounts of vitamin C, which improves iron absorption and availability. The main sources of iron were meat and cold cuts (heme iron) in the CM group, and fortified milk in the YCF group. Similar results were reported in research studies conducted in Sweden [54] and New Zealand [55], where dietary iron intake and the prevalence of anemia (iron deficiency) did not differ between children consuming CM and YCFs. Some studies postulated a negative correlation between the consumption of CM and dietary iron intake. Daily consumption of unmodified CM in excess of 450 g could increase the risk of iron deficiency [56,57].

The tolerable upper intake level (UL) of selected micronutrients, such as vitamin A, could be exceeded in children consuming fortified foods, in particular when the daily consumption of YCFs exceeds 600 mL [12,58]. In the studied population, vitamin A intake exceeded reference levels two-fold in the YCF group, and it reached 132.8% of the guideline value in the CM group, where the main sources of vitamin A were butter, CM, eggs and carrots. In the PITNUTS study, vitamin A requirements were met by 96.6% of the children’s diets [16]. The average dietary intake of vitamin A in children aged 13–24 months was insufficient only in the work of Sharma (2013) [59].

### 4.3. Consumption of Young-Child Formulas and Cow’s Milk vs. Children’s Dietary Preferences

The second year of life is a sensitive period for food acceptance and the development of eating habits. Cow’s milk or YCFs are introduced to expand children’s diets, but little is known about their influence on the formation of eating habits and nutrient intakes in 2-year-olds. Numerous researchers have expressed their concerns regarding YCFs as a substitute for a diverse diet, because children fed YCFs are more likely to become dependent on liquid food than those consuming regular homemade meals, and because YCFs may affect satiation in infants [4,60]. A recent position paper by ESPGHAN postulates that YCFs do not have to be administered routinely but could play a minor role in dietary strategies aiming to increase the intake of nutrients such as iron and vitamin D [4].

In the studied population, the introduction of YCFs significantly increased the consumption of sweetened dairy products, beverages and juice, and decreased the intake of plain fermented milks. There is evidence to suggest that the sweetness of YCFs causes babies to crave sweet taste. Sucrose-containing formulas were also found to boost appetite and promote hypersensitivity to sweetness [61,62,63]. Children consuming CM were less likely to drink sweetened beverages, their daily carbohydrate intake was nearly 20% lower and the percentage of total energy intake from carbohydrates was consistent with dietary guidelines. Contrary results were reported by Lovel who found that consumption of YCFs was associated with higher nutritional adequacy and an increased likelihood of meeting nutrient requirements. However, the impact of the family diet and YCFs on dietary diversity requires further evaluation [11].

### 4.4. Strengths and Limitations

The main strength of the present study was that the diets of children aged 13–24 months were evaluated comprehensively by analyzing the daily intake of various food products as well as the overall quality of infant diets. Dietary and feeding patterns in toddlers (12 months to preschool age) have been insufficiently investigated in the Polish population. The division of the studied population into a group consuming YCFs and a group consuming CM supported the search for associations between milk type and the choice of other food products in a child’s diet. This is an important consideration in view of excess intake of energy, juices and saturated fats, as well as the formation of desirable eating habits. The interviewed parents were eager to cooperate, and a high percentage of the respondents completed the questionnaire and kept food diaries. The respondents were also asked to state the reasons for choosing a given YCF, which was yet another strength of the study.

The limitations of the study include the fact that the parents measured the children’s dietary intakes with the use of household utensils and that foods which were not consumed during a meal were not taken into account in dietary recalls. Homemade meals were also problematic because their energy value was estimated based on meals that are most frequently served to Polish children. The study accounted for differences in the fat content of CM, whereas milk bought from a vending machine (which was listed in the dietary recalls of three children) was regarded as whole milk. The time required to complete the questionnaire was also a limitation. In the future, studies of the type could rely on telephone surveys to save the respondents’ time and minimize direct contact between the respondents and the researchers, which is a particularly important consideration during the COVID-19 pandemic. The aim of the study was to investigate the eating habits of children aged 13–24 months and to identify undesirable behaviors that could negatively affect children’s growth and development. The study did not attempt to identify the relationships between poor dietary habits and the risk of lifestyle diseases in later life, and further research is needed to explore this problem.

## 5. Conclusions

The daily food intake and the quality of the diets administered to children aged 13–24 months were evaluated and compared with the model food ration. The estimated number and size of portions involving products from all food groups were not consistent with nutritional guidelines. Excessive or inadequate consumption of selected products, relative to the model food ration, can increase the risk of lifestyle diseases in later life. Quantitative and qualitative modifications of the consumed diets could improve the nutritional status of the assessed children. The results of the study were also used to identify associations between the type of consumed milk, the nutritional value of children’s diets and deficiency or excess of nutrients that are essential for healthy growth and development. It was found that milk type influenced children’s eating habits and preference for sweet-tasting foods. The study also demonstrated that Polish parents and caregivers only have limited knowledge of nutritional guidelines for toddlers. Most respondents chose milk formulas based on price and availability, rather than nutritional content. The present findings can be used by pediatricians and dieticians to educate parents about the nutritional composition of YCFs and the benefits of reducing the amount of sugar in children’s diets.

## Figures and Tables

**Table 1 nutrients-13-02511-t001:** The estimated number of servings and serving size of different food products in the daily rations of children aged 13–24 months and guideline values [17].

	Guidelines		Dietary Recalls	
	No. of Servings	Serving Size	Mean No. of Servings	Serving Size
I. Protein foods	**4–5 Servings**		**5–6 Servings**	
Dairy foods	3 servings	- 1 cup of milk (2%–3.2% fat content), - 1 cup of modified milk (for children aged 1–3 years), - ½ cup of yogurt, kefir or 2 teaspoons of tvorog, cottage cheese or 1 teaspoon of grated yellow cheese	2–3 servings	- 1 cup of milk (2%–3.2% fat), - 1 cup of modified milk (for children aged 1–3 years), - 1 small tub of yogurt or cream cheese dessert, - 1 teaspoon of grated yellow cheese or a slice of tvorog
Other protein foods	1–2 servings	- ½ egg or 1 slice of cold cuts made of lean meat, - 1 serving of meat or fish	3 servings	- 1 egg/2 slices of ham, - 1 sausage, - 1 serving of roast meat or fish
II. Cereal products	5 servings	- ½ wheat roll or 1 slice of wheat bread, - ½ whole wheat roll or 1 slice of whole wheat bread, - ½ cup of cereal, - 2–3 tablespoons of cooked groats, pasta or rice, - crepe, pancake or cake	4 servings	- 1 wheat roll or 1 slice of wheat bread, - ½–1 cup of breakfast cereal, - 2–3 tablespoons of cooked groats, pasta or rice, - crepe, pancake or cake
III. Vegetables	5 servings	- yellow vegetables, e.g., 5 string beans, - orange vegetables: 2 tablespoons of grated carrots, cooked pumpkin, - white vegetables: ½ cup of cabbage/sauerkraut, ½ cup of cooked cauliflower, - red vegetables: 1 small tomato/ ½ small ball pepper, - green vegetables: lettuce, ½ cup of cooked broccoli or spinach	2–3 servings	- orange vegetables: 2–3 tablespoons of grated carrots, - red vegetables: 1 small tomato/1 cooked beet/2 tablespoons of blanched red cabbage, - green vegetables: lettuce, chives, 2–3 tablespoons of cooked broccoli or courgette
IV. Fruit	4 servings	- yellow and orange fruit: 1 apple, 2–3 apricots, ½ banana or orange, - purple fruit: ½ cup of currants/berries, - red fruit: ½ cup of raspberries or strawberries, - ½ cup of freshly squeezed fruit juice	2–3 servings	- yellow and orange fruit: 1 apple, ½ banana or orange, or a small tangerine, - 1–1½ cups of bottled or freshly squeezed fruit juice
V. Fat	1–2 servings	- 1 tablespoon of olive oil or rapeseed oil - 1 teaspoon of butter	2–3 servings	- 1–2 teaspoons of butter/margarine - teaspoon of rapeseed oil

**Table 2 nutrients-13-02511-t002:** The recommended daily ration for children aged 13–24 months and the mean daily intake of various food products in the analyzed food diaries.

Product Groups	Unit	Daily Ration Based on Guidelines	Mean Daily Ration in Food Diaries	% of the Model Food Ration (2013)
		Model Food Ration (2013)		
Cereal products				
Bread	g	20	25	125
Flour, pasta	g	25	30	120
Groats, rice, breakfast cereal	g	30	42	140
Potatoes	g	80–100	50	50–62.5
Vegetables and fruit	g	450	355	78.9
Vegetables	g	200	140	70
Fruit	g	250	215	86
Milk and dairy products	g			
Milk, modified milk and fermented milks	g	650	360	55.4
Tvorog and cottage cheese	g	10–15	5	33.3–50
Rennet cheese	g	2	10	500
Meat, cold cuts and fish	g			
Red meat and poultry	g	15	20	133.3
Cold cuts	g	5	40	800
Fish	g	10	5	50
Eggs	g/number	½ egg	½ egg	100
Fat	g	16	22	137.5
Animal fat: butter and cream	g	6	20	333.3
Vegetable fat: oil	g	10	2	20
Sugar and sweets	g	20	48	240

**Table 3 nutrients-13-02511-t003:** Nutritional value of YCFs and whole cow’s milk.

Nutritional Value per 100 g of the Product	YCF ^1^ (Min–Max)	SD	Whole Cow’s Milk ^2^
Energy, kcal	70 (65–86)	3.53	61
Fat, g	3.05 (2.6–4.2)	0.58	3.2
Saturated fatty acids, g	0.94 (0.6–1.3)	0.31	1.92
Monounsaturated fatty acids, g	1.37 (0.97–1.9)	0.31	1.01
Polyunsaturated fatty acids, g	0.61 (0.4–1.0)	0.22	0.08
Carbohydrates, g	8.68 (8.1–9.2)	0.44	4.8
Sugar, g	7.47 (6.1–9.0)	1.11	4.8
Lactose, g	6.6 (5.4–9.0)	1.49	4.6
Sucrose, g	0.9 (0.0–2.4)	0.86	0.2
Fiber, g	0.66 (0.32–1.0)	0.27	0
Protein, g	1.64 (1.0–3.2)	0.78	3.3
Sodium, g	24.74 (13.45–32.0)	13.1	44
Vitamin A, μg	61.17 (50.0–69.0)	6.88	36
Vitamin D, μg	2.23 (1.05–3.2)	0.93	0.03
Vitamin E, mg	0.93 (0.8–1.1)	0.12	0.10
Vitamin C, mg	12.61 (8.4–15.0)	2.87	1.0
Thiamine, mg	0.05 (0.04–0.07)	0.01	0.04
Riboflavin, mg	0.20 (0.097–0.3)	0.07	0.17
Niacin, mg	0.58 (0.2–1.2)	0.36	0.1
Vitamin B6, mg	0.08 (0.06–0.105)	0.02	0.05
Folic acid, μg	11.38 (9.0–13.0)	1.36	5.0
Vitamin B12, μg	0.39 (0.2–0.61)	0.16	0.4
Calcium, mg	133.5 (83.0–217.0)	44.4	118
Iron, mg	1.22 (0.8–1.7)	0.29	0.1
Zinc, mg	0.7 (0.4–0.9)	0.18	0.32
Magnesium, mg	11.66 (6.3–20.0)	5.1	12
Iodine, µg	22.03 (13.8–42.0)	10.1	No data

^1^ based on the nutrition facts labels of six YCFs available on the Polish market.^2^ nutritional value of whole cow’s milk according to Polish producers.

**Table 4 nutrients-13-02511-t004:** Energy content nutritional value of the evaluated diets of children aged 13–24 months vs. dietary guidelines and expert recommendations.

No.	Nutrient	Unit	Mean Nutritional Value of the Evaluated Diets ±SD (*n* = 334)	Dietary Guidelines (Jarosz 2020)		Children Receiving YCF (*n* = 178)		Children Receiving Cow’s Milk (*n* = 156)		
				RDA	% RDA	Mean Nutritional Value	Mean Nutritional Value of YCF Diets (Mean Intake—283 mL)	Mean Nutritional Value	Mean Nutritional Value of Cow’s Milk Diets (Mean Intake—261 mL)	*p*-Value
Energy										
1	Energy	kcal	1051.8 ± 155.07	1000	105	1136.7	198.1	1023.4	158	<0.05
Nutrients										
2	Total protein, including animal protein	g	24.6 ± 5.27	14	175.7	21.9	4.6	29.3	8.61	<0.01
3	Total fat, including	g	33.49 ± 6.96	33–44 11.1 *	76.9–101.5	34.26	8.6	32.81	8.4	ns
	saturated fatty acids	g	10.6 ± 3.7		95.5	10.9	2.66	10.4	5.0	ns
	Monounsaturated fatty acids	g	8.6 ± 3.2		-	9.42	3.9	7.23	2.6	<0.01
	Polyunsaturated fatty acids	g	2.2 ± 1.1		-	2.2	1.7	2.2	0.2	ns
5	Total carbohydrates, including:	g	153.18 ± 16.9	130 ** <25	117.8	180.2	24.6	146.9	12.5	<0.01
	Sucrose	g	45.7 ± 8.1		182.8	67.6	2.5	33.8	0.52	<0.001
	Lactose	g	26.3 ± 2.29			27.2	18.7	25.4	12.0	ns
6	Dietary fiber	g	10.5 ± 2.29	10(AI)	105	9.46	1.87	11.3	0	ns
7	Energy from protein	%	9.7	5–15	194–64.6	9.1	-	11.45	-	ns
	Energy from fat	%	29.7	35–40	84.8–74.3	27.1	-	28.86	-	ns
	Energy from carbohydrates,	%	60.6	45–65	134.6–93.2	63.4	-	57.4	-	ns
	including sucrose	%	17.4	<10	174	23.8	-	13.21	-	<0.001
8	Water/beverages	mL	1204.2 ± 179.3	1250 (AI)	96.3	1280.8	283	1120.1	261	<0.05

* Recommended maximum amounts per day; ** Not less than 130 g/day; AI—adequate intake.; ns—not statistically significant.

**Table 5 nutrients-13-02511-t005:** Energy and nutritional value of the evaluated diets of children aged 13–24 months vs. dietary guidelines and expert recommendations.

No.	Nutrient	Unit	Mean Nutritional Value of the Evaluated Diets ±SD (*n* = 334)	Dietary Guidelines (Jarosz 2020)		Children Receiving YCF (*n* = 178)		Children Receiving Cow’s Milk (*n* = 156)		*p*-Value
				RDA	% RDA	Mean Nutritional Value	Mean Nutritional Value of YCF Diets (Mean Intake—230 mL)	Mean Nutritional Value	Mean Nutritional Value of Cow’s Milk Diets (Mean Intake—180 mL)	
Fat-soluble vitamins										
1	Vitamin A	μg	734.7 ± 175.1	400	183.6	926.1	172.4	531.3	93.9	<0.001
2	Vitamin E	mg	6.01 ± 1.4	6(AI)	100	6.1	2.63	5.84	0.26	ns
3	Vitamin D	μg	5.7 ± 1.7	15(AI)	38	9.81	6.3	2.11	0.07	<0.001
Water-soluble vitamins										
4	Vitamin B1	mg	0.44 ± 0.18	0.5	88	0,47	0.14	0.42	0.1	ns
5	Vitamin B2	mg	0.62 ± 0.16	0.5	124	0.72	0.57	0.58	0.45	ns
6	Foliates	μg	138.9 ± 19.4	150	92.6	136.6	26.17	140.3	13.05	ns
7	Vitamin B12	mg	0.9 ± 0.46	0.9	100	0.94	1.1	0.83	1.04	ns
8	Vitamin C	mg	61.5 ± 36.2	40	153.8	86.8	35.7	40.7	2.61	<0.01
10	Vitamin B6	mg	0.43 ± 0.17	0,5	86	0.44	0.22	0.43	0.13	ns
Minerals										
1	Calcium (Ca)	mg	689.57 ± 167.1	700	98.5	714.2	377.7	663.7	310.9	ns
2	Phosphorus (P)	mg	678 ± 186.2	460	147.4	765.4	187.8	619.8	222.9	<0.01
3	Magnesium (Mg)	mg	62.6 ± 15.3	80	78.3	61.7	32.9	63.8	31.3	ns
4	Iron (Fe)	mg	7.45 ± 1.9	7	106.4	7.61	3.46	7.31	0.26	ns
5	Zinc (Zn)	mg	1.8 ± 1.2	3	60	2.43	1.98	1.4	0.84	<0.01
8	Iodine (I)	μg	94.6 ± 17.3	90	105	91.6	28.7	98.3	28.6	ns
9	Potassium (K)	mg	1097.5 ± 277.1	800(AI)	137.2	1190.2	300.2	1009.4	363.8	<0.05
10	Sodium (Na)	mg	957.1 ± 229.9	750(AI)	127.6	1014.2	70.0	946.5	114.8	<0.05

AI—adequate intake; ns—not statistically significant.

**Table 6 nutrients-13-02511-t006:** Odds ratios (95% confidence interval). The associations between the type of consumed milk, YCF vs. cow’s milk, and the composition the diets of children aged 13–24 months.

Product	YCF Group vs. Mixed Group			Cow’s Milk Group vs. Mixed Group		
	OR	95%CI	* *p*-Value	OR	95%CI	*p*-Value
Juice	1.43	0.91–2.24	**	0.94	0.73–1.21	ns
Sweetened beverages	2.54	1.32–3.26	***	0.79	0.56–1.08	ns
Sweet snacks	0.76	0.44–0.94	**	0.91	0.73–1.17	ns
Sweets	1.17	0.86–1.32	ns	0.97	0.79–1.10	ns
Natural dairy products, plain yogurt	0.56	0.21–0.79	***	1.84	1.26–2.41	**
Sweetened dairy products, such as cream cheese dessert	1.87	1.36–2.54	**	0.64	0.26–0.89	***
Breakfast cereal	0.9	0.86–1.19	ns	1.28	0.79–1.56	*
Fruit preserves	1.17	0.86–1.32	ns	1.06	0.76–1.14	ns

Statistically significant (Wald’s statistics): * *p* value < 0.05, ** *p* < 0.01, *** *p* < 0.001, ns—not statistically significant.

## Data Availability

Due to ethical restrictions and participant confidentiality, data cannot be made publicly available. All data were collected in an anonymous way. The data that support the findings of this study are available from the authors upon reasonable request the corresponding author and with permission by main author.

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
