# Peer review of "A Comparison of the Effects of Young-Child Formulas and Cow’s Milk on Nutrient Intakes in Polish Children Aged 13–24 Months"

_nutrients, 2021, doi:10.3390/nu13082511_

Round 1

Reviewer 1 Report

This is a prospective cohort study comparing toddlers who consuming cow’s milk (CM) or young-child formula (YCF). This article is well written. I have several comments here.

  1. In Table 2, although AAP guideline was listed, there is no comparison ration for it. It is hard to understand what the authors mean to list AAP guideline value.
  2. From Table 1 to Table 3, they have nothing to do with the conclusion of this article. However, the authors used a lot of words to describe them. Those results can be reported in a simpler way.
  3. Page 14, line 333, the authors described that “Children fed YCF at the age of 2 years....”. It is not compatible with the title of table 6. (Aged 13-24 months should be “age of 1 year”)
  4. In table 6, the odds ratio and p value must have something wrong. For example, “Sweet snacks” has a odds ratio of 0.44~0.94. However, the p value is 0.32. It is very unreasonable. Please consult statisticians for odds ratio calculation because the only conclusion of this article comes from Table 6.
  5. It is a pity that this article did not list the most important outcomes for children’s diet. That is growth and development. I think this is a major drawback for this research.

Author Response

Thank you very much for your insightful review. We greatly appreciate the time and efforts to review our manuscript and we agree that the proposed changes will contribute to the improvement of our manuscript. We hope you will find our improvements appropriate and comprehensive.

In Table 2, although AAP guideline was listed, there is no comparison ration for it. It is hard to understand what the authors mean to list AAP guideline value.

Thank you very much for your comment and paying attention to the structure of table 2. The AAP guidelines have been included in the table to show the differences from the model food ration. But there is indeed no reference to these guidelines and they have been removed from the table.

From Table 1 to Table 3, they have nothing to do with the conclusion of this article. However, the authors used a lot of words to describe them. Those results can be reported in a simpler way.

Thank you for your comment. Following the reviewer's recommendation, the description of the results in Tables 1 and 2 was slightly shortened (line 224 and line 242 and line 272) and a short comment was added to the conclusions based on the results presented in these tables (lines 555-562). Table 3 seems important to the authors due to the presentation of the average nutritional value of milk available on the Polish market.

Page 14, line 333, the authors described that “Children fed YCF at the age of 2 years....”. It is not compatible with the title of table 6. (Aged 13-24 months should be “age of 1 year”)

Thank you for your attention, the wording used to describe the table has been corrected and standardized for children aged 13-24 months (line 345).

The previously used term for children at the age of 2 years is equivalent to the period in question in Polish. But in order not to mislead the reader, this has been corrected.

In table 6, the odds ratio and p value must have something wrong. For example, “Sweet snacks” has a odds ratio of 0.44~0.94. However, the p value is 0.32. It is very unreasonable. Please consult statisticians for odds ratio calculation because the only conclusion of this article comes from Table 6.

Thank you for your attention, indeed, an editorial error appeared while completing the data for publication. We are very sorry for your inconvenience, the table has been recalculated.

It is a pity that this article did not list the most important outcomes for children’s diet. That is growth and development. I think this is a major drawback for this research.

Thank you for your comment. Indeed, a reference to the health and normal development of children would be a valuable contribution to the publication. But that was not the purpose of the study. In the Strength and limitation section, this is described (lines 547-552). In the discussion, we extended the reference of the obtained results, especially with regard to nutritional deficiencies or excesses, and emphasized their possible negative impact on the health and development of children.

Lines; 417-420; 429-435; 467.

Reviewer 2 Report

Original paper, prepared by authors who are very competent in the field of rational nutrition of Polish children and adolescents. The publication complements the scientific achievements of other Polish authors dealing with the nutrition of the population of children, adolescents and adults in health and disease, as evidenced by the citations (No. 17-21, 31). Researchers, using the survey method (three-day questionnaire), obtained a lot of valuable information from parents of healthy Polish children, aged 13-24 months, on the qualitative and quantitative composition of their daily diet. The obtained data concern a representative group of Polish children. The results of the research confirm large discrepancies between the recommendations of Polish guidelines and international guidelines regarding the principles of nutrition in this age group. These discrepancies are fully discussed in the Discussion chapter. The reviewer shares the authors' expectations that present findings can be used by paediatricians and dietitians to educate parents about the nutritional composition of cow’s milk products and YCFs, and the benefits of reduction the amount of sugar and other ingredients in children's diets.

Author Response

Original paper, prepared by authors who are very competent in the field of rational nutrition of Polish children and adolescents. The publication complements the scientific achievements of other Polish authors dealing with the nutrition of the population of children, adolescents and adults in health and disease, as evidenced by the citations (No. 17-21, 31). Researchers, using the survey method (three-day questionnaire), obtained a lot of valuable information from parents of healthy Polish children, aged 13-24 months, on the qualitative and quantitative composition of their daily diet. The obtained data concern a representative group of Polish children. The results of the research confirm large discrepancies between the recommendations of Polish guidelines and international guidelines regarding the principles of nutrition in this age group. These discrepancies are fully discussed in the Discussion chapter. The reviewer shares the authors' expectations that present findings can be used by paediatricians and dietitians to educate parents about the nutritional composition of cow’s milk products and YCFs, and the benefits of reduction the amount of sugar and other ingredients in children's diets.

Thank you very much for your insightful review. We also hope that the presented work will complement the knowledge about the nutrition of children aged 13-24 months in Poland. Each such study has a chance to contribute to the promotion of proper nutrition and to learn about errors and problems that can be corrected. The work has been checked by a native speaker.
